# The Impact of Scale on Extracting Individual Mobility Patterns from Location-Based Social Media

**DOI:** 10.3390/s24123796

**Published:** 2024-06-12

**Authors:** Khan Mortuza Bin Asad, Yihong Yuan

**Affiliations:** Department of Geography and Environmental Studies, Texas State University, San Marcos, TX 78666, USA; k_b501@txstate.edu

**Keywords:** geospatial big data, scale, spatial scale, location-based social media data, human movement, urban mobility

## Abstract

Understanding human movement patterns is crucial for comprehending how a city functions. It is also important for city planners and policymakers to create more efficient plans and policies for urban areas. Traditionally, human movement patterns were analyzed using origin–destination surveys, travel diaries, and other methods. Now, these patterns can be identified from various geospatial big data sources, such as mobile phone data, floating car data, and location-based social media (LBSM) data. These extensive datasets primarily identify individual or collective human movement patterns. However, the impact of spatial scale on the analysis of human movement patterns from these large geospatial data sources has not been sufficiently studied. Changes in spatial scale can significantly affect the results when calculating human movement patterns from these data. In this study, we utilized Weibo datasets for three different cities in China including Beijing, Guangzhou, and Shanghai. We aimed to identify the effect of different spatial scales on individual human movement patterns as calculated from LBSM data. For our analysis, we employed two indicators as follows: an external activity space indicator, the radius of gyration (ROG), and an internal activity space indicator, entropy. These indicators were chosen based on previous studies demonstrating their efficiency in analyzing sparse datasets like LBSM data. Additionally, we used two different ranges of spatial scales—10–100 m and 100–3000 m—to illustrate changes in individual activity space at both fine and coarse spatial scales. Our results indicate that although the ROG values show an overall increasing trend and the entropy values show an overall decreasing trend with the increase in spatial scale size, different local factors influence the ROG and entropy values at both finer and coarser scales. These findings will help to comprehend the dynamics of human movement across different scales. Such insights are invaluable for enhancing overall urban mobility and optimizing transportation systems.

## 1. Introduction

Human activities and movements significantly influence and shape an urban system. Identifying human movement patterns in an urban area has many applications, including transportation planning and traffic forecasting, urban planning, public health and epidemiology, emergency management, etc. [1,2]. Because of the recent advancements in location-aware devices, GPS systems, data handling, and data storage technologies, there is a large number of spatiotemporal datasets that can be used to study human movement patterns [3,4]. Most previous studies on human movement patterns from big geodata focused on modeling either individual or collective human movement patterns in either spatial or temporal dimensions [5,6,7,8]. Individual movement pattern modeling provides information about how an individual travels in space and time [1,9,10]. This category of research serves as a valuable tool for capturing the dynamic patterns of human mobility across various spatial and temporal scales, while also exploring the interplay between individuals and their environments. Aggregated human mobility modeling offers insights into the mobility patterns of an entire population within a specific area over a given timeframe [1,9,10]. Such aggregated movement modeling can help gain a deeper understanding of the dynamic and intricate urban systems and activities in various regions of a city as they evolve over time. Although both individual and aggregated human movement pattern identification is important to detect the overall human movement patterns in space and time efficiently, very few studies have focused on the effect of spatial scale on human movement pattern identification from geospatial big data.

Within the field of individual activity studies, the investigation of activity spaces holds a significant place. In the previous literature, activity space is defined as the area within which people travel to perform their daily activities [11,12,13]. Different indicators have been used to model the external morphology (e.g., shape and size) and internal structure (patterns of visiting different points of interest) of an activity space [14,15]. Previous studies show that the radius of gyration (ROG) and entropy are the two commonly used indicators to model the external morphology and internal structure of an activity space, respectively [15,16]. ROG is less susceptible to outliers and entropy can capture the randomness of user activity patterns, which are very crucial for sparse datasets like location-based social media (LBSM) data [14]. While both the ROG and entropy are widely utilized in modeling human activity spaces, there is a notable gap in the research concerning quantifying how different spatial scales affect activity space modeling, particularly when dealing with prevalent big data sources like LBSM data in the digital age [17,18]. Varying spatial scales can significantly impact the interpretation of various activity space indicators, particularly during the data preprocessing stage. For instance, consider two activity points of the same user located just 10 m apart. In such cases, it might be appropriate to group them into a single point, as they likely represent the same location, such as a user being indoors in the living room or outdoors in the backyard at home. However, this process requires careful consideration, as aggregating points can result in the loss of important information when calculating activity space indicators.

In this study, we aim to address this existing knowledge gap by investigating how different grid sizes, used to group activity points, may affect activity space indicators. To conduct this analysis, we use Weibo check-in data, which is the Chinese equivalent of Twitter. To understand the impact of spatial scale on activity spaces, we divide our study area into various grid sizes, where nearby activity points are grouped within the same grid. Subsequently, we calculate ROG and entropy values using different grid sizes for three Chinese cities. The central research question we seek to answer is how changes in spatial scale, as represented by grid size, influence the identification of individual movement patterns, especially when using sparse datasets like LBSM data.

## 2. Literature Review

### 2.1. Modeling Human Activity Space

Modeling human activity space is important to study individual travel patterns. In urban geography, activity space is known as the local area in which people travel to perform their daily activities [11,12,13]. Horton and Reynolds (1971) defined activity space as “an individual’s activity space is defined as the subset of all urban locations with which the individual has direct contact as the result of day-to-day activities” (p. 37) [19]. Many researchers focused on the spatial and temporal aspects of activity space [7,20,21]. Some other researchers focused on different measurement dimensions, for example, the extent, intensity, diversity, and exclusivity of activity space [22,23]. The spatial measurement of activity space has evolved with different data sources, and many researchers have grouped it into three categories as follows: point, distance, and area [22,24,25]. Point measures usually comprise nodes [24] and kernel density of activities [26,27]. Distance indicators usually comprise trip length [28], the radius of gyration [29], and the space–time path [23]. Areal measures usually comprise minimum convex polygons [30] and standard deviational ellipses [31,32]. Many of these methods have been used in previous studies to model human activity space using traditional transportation data sources collected from questionnaire surveys, travel diaries, etc. However, these traditional data are sometimes costly to collect, covering a small sample of the population, which makes it difficult to interpolate for the whole population of the study area, and they are also difficult to collect for a long period of time [33,34,35]. Therefore, researchers have started using different big geospatial data sources to model human activity space in recent times. Moreover, the improvement in data collection, storage, and handling technologies has made it easy to use these big geospatial data [36,37]. In this big data era, LBSM data are being widely used for modeling human activity patterns [17,38]. Although these big data sources have some quality issues [14,15,39], they are very popular among current researchers as they are easy to collect.

Previous studies have reconstructed individuals’ frequently traveled routes by representing these paths as sequences of longitude and latitude coordinates through sequential pattern mining or trajectory mining algorithms [40,41]. When assessing user mobility behaviors, interpreting raw trajectories can be challenging. This difficulty arises because it is hard to correlate specific points within these trajectories to places of notable interest, such as landmarks, downtown areas, sports or concert events, or museums [42]. To address this issue, these trajectories are frequently transformed into sequences representing regions of interest, thereby providing a clearer and more meaningful analysis of movement patterns [43]. Utilizing big geodata sourced from social media significantly complicates the knowledge extraction process. This complexity stems from the necessity of employing scalable, parallel, and distributed techniques to handle and analyze the vast volumes of data involved efficiently [44,45]. These advanced methods are crucial to managing the scale and intricacy of social media data, as well as ensuring meaningful insights can be derived when handling data with inherent uncertainties [46]. As these data often contain inherent uncertainties, such as inaccuracies or inconsistencies in user-generated content, it becomes essential to employ robust analytical frameworks.

Similar to trajectory analysis, researchers need to be careful while using big geospatial data to model human activity space because of potential data quality issues (e.g., low resolution, incompleteness, and inconsistency) [14,15,39]. Yuan and Wang [15] performed a thorough analysis of seven commonly used activity space indicators that are used to model human activity space and evaluated their performances for LBSM data. Four of the indicators were chosen in that study to measure the external morphology of an activity space (alpha shape, minimum convex hull, standard deviational ellipse, and radius of gyration), and three of them were chosen to measure the internal structure (home location to represent the most important point of interest, minimum spanning trees, entropy). They found that ROG and entropy are more stable indicators than the others and seem to be less affected by the size of the activity space and the sparseness of data [14]. Table 1 shows a comparison of the pros and cons of different activity space indicators. In this study, we attempt to identify the effect of spatial scale on human movement patterns in urban areas. We used Weibo data for this study, which are LBSM data. We chose ROG and entropy as the activity space modeling indicators in this study for the aforementioned reason.

### 2.2. The Effect of Spatial Scale on Human Mobility Patterns

In the previous literature, there are two major groups of studies on human movement patterns including individual movement patterns and aggregated/collective movement patterns [4,47]. In both scenarios, it is important to consider the specific spatial and temporal scales inherent to mobility, which can range from mere hundreds of meters to thousands of kilometers and from hours to years [4]. Different modeling frameworks have been used for each of these cases [4,47,48]. For example, various modifications of the “Random Walk” model have been used for individual-level mobility, while the “Gravity Model”, “The Radiation Model”, and others have been used for population-level mobility modeling. In this study, we focus on the effect of different spatial scales on individual-level mobility patterns. Individual mobility exhibits a degree of uncertainty linked to personal choices and arbitrary actions, introducing an element of randomness into travel patterns. However, various studies have emphasized that individual trajectories are anything but random; they often display a high degree of regularity and predictability [4]. This predictability can be harnessed to forecast an individual’s future locations and construct models of individual mobility, as well as provide informative input for urban planners and policymakers. For example, mobility studies in smart cities play a crucial role in public health by providing insights into population movement patterns and their implications on the spread of diseases. By analyzing data from various sources, including GPS, mobile phones, and social media, researchers can identify high-traffic areas and potential hotspots for disease transmission [49]. This information is invaluable for designing targeted interventions, optimizing resource allocation, and enhancing the overall effectiveness of public health responses. For instance, during the COVID-19 pandemic, mobility data were essential in predicting outbreak locations and evaluating the impact of lockdown measures [50,51].

The concept of spatial scale has been explored extensively and categorized into four different categories in geography [52,53,54]. These categories are cartographic, observational, measurement, and operational. The cartographic scale is used to demonstrate the relationship between the size of map features and their actual dimensions on the ground [52,55]. The observational scale delineates the extent of the area under investigation [52]. The measurement scale, synonymous with resolution, signifies the smallest discernible unit within a study [56]. The choice of the measurement scale significantly impacts both the input data and the outcomes of analysis because they change depending on the size and shape of the area unit of analysis [52,55,56]. Lastly, the operational scale typically denotes the spatial scope within which specific environmental processes actively occur [52]. It represents the level at which the greatest range of pattern variability can be effectively observed and analyzed. The operational scale plays a decisive role in determining the dimensions of spatial coverage (observational scale) and the level of detail (measurement scale) [52,55].

Nevertheless, very limited research has been conducted to investigate the impact of spatial scale on the identification of individual mobility patterns. This is a significant gap to address because the existing studies have demonstrated that relationships established at one spatial scale do not necessarily hold when applied to other spatial scales in a linear fashion [57,58,59,60]. Furthermore, relying solely on analysis at a single spatial scale does not provide a comprehensive understanding of the actual spatial patterns [61,62]. To address this issue, some researchers have adopted multi-scale models that aggregate data at various scales [63,64,65]. However, this approach has primarily been used for data management, storage, visualization, and sharing, and not for the exploration of spatial patterns because of its computational complexity [36]. For urban-oriented studies, different sizes of regular grids are commonly employed, ranging from 200 m [66] and 250 m [48] to 500 m [67] and 1000 m [1]. To facilitate comparison across different studies, we choose a grid-based division of the study area instead of density-based. We try to fill the research gap of how different grid sizes (measurement scales) affect individual human movement pattern identification. We aim to explore the effect of spatial scale on individual movement patterns by changing the size of the measurement scale. We use different-sized regular grids to divide Weibo check-in data and investigate how different grid sizes influence the individual activity space indicators extracted from the data.

## 3. Data and Methods

### 3.1. Data

As mentioned in Section 1, in this study, we use Weibo check-in points that were collected for three highly developed cities of China, namely, Beijing, Shanghai, and Guangzhou. Beijing is the capital of China with a population of 21.7 million according to the 2016 census data. Shanghai and Guangzhou are two important port cities in China with populations of 24.15 million and 12.70 million, respectively [68]. We only included the population within the city limit instead of the population of the entire metropolitan area. A total of 1.18 million georeferenced Weibo check-in points were collected from April 2015 to March 2016 for all three cities. Among all the fields extracted from Sina Weibo, we only used three fields as our data attributes including the unique identifier (i.e., user account ID), the coordinates of check-in locations, and the timestamp of check-ins. The official Weibo API was used to collect the data. Only the users who had at least 10 check-in points were included to ensure that there was enough information about an individual user’s activity pattern. This approach helped us to address the issue of data sparseness by excluding potential short-term visitors or inactive users of LBSM during the study period. Table 2 shows the information on the collected Weibo data after filtering and cleaning.

### 3.2. The Distribution of Data in Different-Sized Grids

The activity space of an individual is the area within which that person performs his regular activities. While calculating activity space indicators, it is often necessary to group nearby points, and it is challenging to determine under what spatial scale this grouping of points should be conducted. On the one hand, if the grid size is too large, one cell of a grid may capture a lot of points and too many points are grouped together. On the other hand, many grid cells might capture no points or very few, resulting in sparse data. Both scenarios can lead to inaccuracies in the calculation of individual activity space indicators.

Here, we use different sizes of spatial grids to group check-in points before calculating the ROG and entropy and investigating how the size of scale (grid size) impacts the values of internal (entropy) and external (ROG) indicators of individual activity spaces. We divide the study areas into grids, ranging from 10 to 100 m with an increment of 10 m, and 100 to 3000 m with an increment of 100 m. The reason why we divide the study area into two different scale ranges is to understand the changes in indicators in a finer spatial resolution and a coarser spatial resolution. We used finer spatial grids (10–100 m) to capture the possible movement of Weibo users at the same place. For example, a user can post on social media from the bedroom, living room, or dining area of their home. These slight changes in location can influence the overall activity space modeling. On the other hand, we use coarser spatial grids (100–3000 m) to capture the movements of a Weibo user in a larger area and among different points of interest. For example, a user can post from their residence, office, shopping center, restaurant, friend or relative’s house, etc., which are far away from each other. Researchers may still choose to model user activities with different levels of detail in this case, which is what the coarser changing grid sizes (i.e., from 100–3000 m) aim to capture. Figure 1 shows the distribution of Weibo check-in points in three different Chinese cities including Beijing, Shanghai, and Guangzhou.

To identify the effect of different scale sizes on individual mobility pattern identification, we divided the study area (all three cities) into different grid sizes. The Weibo check-in points captured by each grid cell are transformed into the middle of the cell. So, it is assumed that the user check-in points captured by a grid cell are in the middle of that grid cell. Figure 2 shows how the distribution of Weibo check-in points changes with the change in grid sizes in Beijing. The other two cities also show a similar pattern.

Figure 3 illustrates the spatial distribution of check-in points within Beijing using two different grid sizes of 1000 m and 3000 m. The finer grid (1000 m) shows a more detailed localization of check-ins, indicating more precise areas of high activity and also showing many areas with sparse data. The larger grid size (3000 m) aggregates the data over a broader area, resulting in less detail but highlighting larger hotspots of activity more clearly. This example shows how the choice of grid size can affect the visibility and interpretation of data concentrations.

### 3.3. Calculating Activity Space Indicators

To model an activity space, it is important to identify its external shape and internal structure. A comparative study of different external (e.g., minimum convex hull, alpha shape, standard deviation ellipse, ROG) and internal (e.g., entropy, minimum spanning tree, kernel density) activity space indicators was performed in [14]. They discovered that among the external activity space indicators, the ROG is less susceptible to outliers. In contrast, among internal activity space indicators, entropy is frequently utilized in prior research to quantify the randomness of user activity patterns, especially when derived from sparse datasets of LBSM data [16]. Therefore, we applied the ROG to model the external shape and entropy to model the internal structure of an activity space in this study. These methods are described in the following paragraphs.

ROG: The radius of gyration is a concept in mechanical engineering, where it is used to identify how the mass of an object is distributed around its mass center. For activity space modeling, ROG is used to identify how points are distributed around the mean center [3,14,69]. In the following equation, *n* refers to the number of points, *r_i_* is the geographic coordinates of each point in the activity space, and *r_m_* is the centroid of all points.


$$ROG=\sqrt{\frac{1}{n}\sum_{i-1}^{n}{\left(r_{i}-r_{m}\right)}^{2}}$$


Entropy: Entropy represents the randomness of activity patterns in the activity space. Higher entropy means more randomness in the activity space. If the points in an activity space are more dispersed, it shows high entropy. On the other hand, if the points are clustered together, it shows low entropy [14]. Also, entropy is relatively stable across different sizes of activity spaces than other internal indicators. Entropy can be computed by the following formula, where *p_i_* is the probability of a point being in the same place *i*, and *N* is the total number of points.


$$E=-\sum_{i=1}^{N}p_{i}{log}_{2}p_{i}$$


First, we divide the study area into grids of varying sizes and then relocate each point to the center of the grid it falls within. We calculate both the ROG and entropy using these centralized locations. The calculations for the ROG and entropy are carried out for grid sizes ranging from 10 m to 100 m in 10 m increments, and from 100 m to 3000 m in 100 m increments. The rationale for employing these two distinct grid size ranges was explained in the previous section. Grid sizes from 10 m to 100 m are intended to capture movements within a small area, sometimes even within the same point of interest. In contrast, grid sizes ranging from 100 m to 3000 m are designed to capture the movements of a Weibo user over larger areas and among various points of interest. A flow chart of the methods used in this study is shown in Figure 4.

In Figure 4, we can see that the first step was to collect Weibo data by using a web crawler and then store it in an offline database. After that, we cleaned and filtered the data to obtain check-ins of the Weibo users in the three study areas (Beijing, Guangzhou, and Shanghai). The next step was to divide the study area into different grid sizes. As mentioned previously in this section, we divided the study area into two different scales and then used the middle point of each cell to represent the locations of the check-in points that fell in the given grid cell. Finally, we calculated the ROG and entropy values of each user with different grid sizes. We used these results to explore the effects of different sizes of scale human movement patterns when extracted from LBSM data.

## 4. Results

### 4.1. Radius of Gyration (ROG)

The ROG is used to identify how the check-in points of a user are distributed around the mean center. A higher ROG value typically indicates that the user traveled greater distances from the activity’s central point over the data collection period, whereas a lower ROG value suggests more confined movements within a smaller radius from this central point. Figure 5 displays the ROG values for grid sizes ranging from 10 m to 100 m, in 10 m increments, across the three studied cities in China. It reveals an overall increasing trend in ROG values as the grid size increases (from 10 m to 100 m) in all three cities. Additionally, Figure 5 presents the R-squared values and regression lines, indicated by dotted orange lines, for the ROG values in Beijing, Guangzhou, and Shanghai. The R-squared values are 0.99 for Beijing, 0.90 for Guangzhou, and 0.95 for Shanghai. These high R-squared values indicate a strong positive correlation between changes in ROG values and grid sizes across all three cities.

We used this relatively small range of grid sizes to capture individual movement patterns in a limited area (e.g., in a house, office, shopping center, etc.). Interestingly, all three cities show ups and downs in the ROG values with an increase in scale size while showing an overall increasing trend (Figure 4). This phenomenon can be explained by the fact that, at certain grid sizes, Weibo check-in points originating from the same point of interest, such as a house, office, or shopping center within the respective cities, tend to be grouped together, effectively appearing as a single point in the grid. Consequently, the ROG value experiences a decrease. Figure 6 shows an example, where the Weibo check-in point distribution of an individual for different grid sizes in Beijing, China. We can see that with the increasing grid size, the check-in points that are close to each other tend to merge and appear as a single point. For the example shown in Figure 6, when the grid size increases from 60 m to 90 m, the two points that are shown in the small box merge and appear as one point. We can see that although, in general, the ROG values increase with the increase in grid size, the ROG value decreases (from 2117.23 m to 2111.45 m) when the grid size increases from 60 m to 70 m.

The variation in the *x*-coefficients across Beijing (1.05), Guangzhou (0.46), and Shanghai (1.03) primarily reflects the differing extents of these cities. A higher *x*-coefficient suggests that ROG values are more responsive to changes in grid size, indicative of a city’s expansive nature with a consistent pattern of urban development. However, a lower *x*-coefficient, as seen in Guangzhou, may imply a smaller city size or a city where ROG values are less influenced by the scaling of the grid size, possibly because of a more intricate blend of urban and natural landscapes within the city’s confines.

Regarding the differences in the R-squared values, these disparities can largely be attributed to the intricacies of each city’s urban structure. A higher R-squared value, like that for Beijing (0.99), signifies a strong linear correlation between grid size and ROG values, which could denote a more homogeneous urban layout. On the other hand, a lower R-squared value, such as Guangzhou’s (0.90), suggests that other factors, beyond just grid size, play a role in determining ROG values, pointing to a more heterogeneous urban environment with a variety of elements influencing its structure, such as the Pearl River, which creates areas of disparate urban density. The irregularity in the increase in ROG values with grid size might be due to the presence of water bodies, parks, or other non-urban areas within the grid that would not contribute to ROG values as much as urbanized areas do.

This research finding can be useful for future studies, as it demonstrates that the ROG value may decrease at a certain scale size when measured in finer detail (10–100 m in the previous case) in urban areas. This decrease happens because social media check-in points within the same building or point of interest tend to merge at larger scales. Researchers can use this method to decide on the spatial scale of their study based on their objectives. For instance, if the aim is to consolidate close social media check-in points (often from the same building or location), a larger grid size is appropriate. If the aim is to keep more detailed data, then a finer grid size is better. For example, Figure 6a shows three check-in points within a 10 m grid. However, as the grid size increases to 90 m, as seen in Figure 6d, the number of check-in points reduces to just one for the same area. Moreover, the geographic features can have a significant effect on the ROG values of that area.

For coarser spatial scales (larger grid sizes ranging from 100 m to 3000 m), we can see that the ROG values keep increasing with the increase in the grid size (Figure 7). This phenomenon occurs because check-in points within the same grid cell group together and centralize within the cell. As the grid size increases, these points tend to diverge since the central points of the larger grid cells are further apart. Consequently, the overall radius of gyration (ROG) value tends to increase. However, there are exceptions. For instance, Figure 8 illustrates a user’s ROG value that decreases when the grid size changes from 2000 m to 3000 m. This indicates that while the general trend is for the ROG value to increase with grid size, it can also decrease. This decrease can happen when check-in points, upon aligning to the center of a larger grid cell, end up closer to each other.

Also, we can see the size of activity spaces (ROG values) for all scale sizes, where Beijing is larger than Shanghai, and Guangzhou is the smallest of the three. This is consistent with the sizes of the cities. Also, similar to the trend shown at the smaller scale of 10–100 m, all cities show a strong linear relationship between grid size and ROG values with an R-squared value of over 0.95, which is indicative of a consistent trend across different urban areas. However, if we compare Figure 7 to Figure 5, we can see that the increases in ROG values with an increase in spatial scale are more consistent with coarser scale sizes than finer scale sizes for Guangzhou and Shanghai, and all three cities show a very high R-squared value (0.98). The lower influence of non-urban elements like water bodies and parks on ROG values could be due to the increased grid size, which reduces the likelihood of a grid cell being entirely within such an area. Consequently, most cells contain at least some Weibo check-in points. Additionally, the slightly greater x coefficient for Beijing (0.40) compared with Guangzhou (0.20) and Shanghai (0.16) suggests that the ROG values in Beijing are more sensitive to changes in grid size than in the other two cities, a difference that might also reflect variations in the overall size of these cities.

The results in this section can help urban planners to understand how residents interact with the urban environment on a macro scale. The decrease in ROG values from a grid size of 2000 m to 3000 m suggests that there can be anomalies in how people’s check-in points are distributed. Urban planners can investigate these anomalies to understand specific local factors that might influence movement patterns, such as the presence of a major transportation hub or a central business district. By observing the check-in points grouping at larger grid sizes, planners can also identify areas within the city that are either under-utilized or over-utilized. Noticing that Beijing has larger activity spaces than Shanghai and Guangzhou, planners can compare urban policies and infrastructure among these cities to understand what influences the size of activity spaces and apply best practices accordingly.

### 4.2. Entropy

The entropy value indicates the randomness of activity patterns in the activity space. Entropy provides insights into how likely a user is to return to previously visited locations and can help predict future trips [13,70]. In this study, a high entropy value suggests that a Weibo user’s check-in points are more dispersed, meaning the user visits a wider variety of places. Conversely, a lower entropy value indicates more regular visiting patterns for the user. Figure 8 displays the average entropy values for grid sizes ranging from 10 m to 100 m, with an increment of 10 m, across all three cities. It shows that the entropy value decreases as the window size increases because closer points merge and appear as a single point in the middle of the grid cell at larger grid sizes. Therefore, the location of check-in points seems less random with an increase in window size. This discovery offers practical value for future research, as it suggests that researchers should anticipate a reduction in entropy values when increasing the cell sizes of small-scale spatial analyses. The entropy values depicted in Figure 8 closely follow an exponential decay curve, illustrating that entropy decreases at a rate proportional to its current value—a characteristic indicated by a constant ratio. This contrasts with linear decay, where the decrease is consistent across each unit. Unlike the linear model used for the radius of gyration (ROG), an exponential decay function was required to aptly model the entropy values. In Figure 8, the blue line represents the actual entropy values, while the orange dashed line indicates the fitted exponential decay model. The exponential decay model is particularly effective in capturing the rapid reduction in entropy values as grid size increases. The exponential decay function is commonly used in different fields to model when a value decreases rapidly with an increase in another value [71]. In this case, the entropy values decrease rapidly with the increase in grid size. Thus, we used the exponential decay function to model this relationship. This relationship is clearly demonstrated by the high R-squared values of 0.99 for Beijing, 1.00 for Guangzhou, and 1.00 for Shanghai. These values signify a very strong correlation between the decrease in entropy and the escalation in spatial scale, which suggests that as the grid size expands, there is a consistent and rapid consolidation of check-in points within the larger grid cells, leading to lower entropy values. Lower entropy in this context implies that check-ins become less dispersed and more predictable as the grid size increases, which may have implications for urban planning and the analysis of human mobility patterns.

The exponential decay model shown in Figure 8 and Figure 9 can generally be represented by the following equation [71,72]:$$y=a*e^{-b.x}+c$$
where y is the dependent variable, representing the quantity undergoing decay at a given value of *x*. In this case, it represents the entropy value, which decreases in response to varying grid sizes.

a is the initial or starting value of the quantity. It represents the value of the quantity at x = 0. In this case, it represents the entropy value if the grid size is 0. But practically, there would not be any check-in points in a grid cell if the grid size is zero [71,72].

e is a mathematical constant approximately equal to 2.71828. It is the base of the natural logarithm and is used in the exponential decay formula.

b is the decay rate or the rate at which the quantity decreases over x. In this case, a larger *b* value indicates a faster decay in entropy values. The negative sign ensures that the exponent is negative, representing decay [71,72].

c is an optional constant and represents a vertical shift in the decay curve. The differences in *c* values suggest that as the grid size becomes very large, the limit to which entropy can decrease varies. In this case, we do not have the parameter *c* because if *x* (size of the scale) increases to infinity, the *y* (entropy) value would be zero. That means when the scale size is so big that the entire study area falls within a very large grid cell, all the check-in points would merge together and appear at the middle of that very large grid cell, and therefore, there would be no variability, and the entropy value would be zero. Hence, we set *c* equal to 0 for this study, and the exponential decay model equation appears as follows:$$y=a*e^{-b.x}$$

Adjusting the parameters of the model allows a user to model different decay processes. The parameters are described in Table 2 to provide a comparative analysis of the three cities for a finer (10–100 m) spatial scale.

In Table 3, we can see that Guangzhou has a higher *a* value than Beijing and Shanghai for small grid sizes (10–100 m). This means, according to the model, that Guangzhou has the highest initial entropy value among all three cities. One possible explanation is that Beijing and Shanghai have more uniform urban development than Guangzhou as the Pearl River divides the city into different sections. Therefore, some grid cells geographically capture parts of both water bodies and land, while others encompass entirely water or land areas. Consequently, the Weibo check-in points exhibit high variability.

All three cities show the same *b* value. That means the rates at which the entropy value decreases with the increase in the grid size are similar for all three cities. This uniformity in the *b* value could mean that factors influencing the entropy decay when the grid sizes are small (10–100 m) acted similarly in all three cities. For example, check-in points at the neighborhood/subdivision level may show comparable densities in the three cities. However, it is essential to interpret this finding within the broader context of the study and consider other parameters and factors as well.

Figure 10 shows the entropy values for grid sizes of 100 m to 3000 m with an increment of 100 m for all three cities. The entropy values decrease with the increase in grid size. When the grid sizes are small, the check-in points appear more dispersed as they show up almost at their original location, and the data show more randomness, but when the grid size increases, many points are aggregated in the bigger grid cells and clustered together in the middle of those bigger cells. For this, the points appear more clustered when the grid size increases, the data show less randomness, and the entropy value drops. So, the calculated randomness or the probability of a user posting on social media from different locations decreases with the increase in the size of the spatial scale. The blue curve shows the entropy values for all three cities and the orange curve shows the exponential decay curve in Figure 10. The high R-squared value in Figure 10 shows that the decrease in entropy values is highly associated with the increase in window size, and it decreases exponentially with the increase in window size (spatial scale).

The parameters are described in Table 4 to provide a comparative analysis of three cities for a coarser (100–3000 m) spatial scale.

In Table 4, we can see that Beijing has lower *a* value than Guangzhou and Shanghai for a large grid size. Both Guangzhou and Shanghai have a major water body that runs through the city and provides some variety in the urban infrastructure. For that reason, the Weibo check-in captured by the grid cells in Guangzhou and Shanghai experienced relatively more variability.

Guangzhou has a higher *b* value than the other two cities, suggesting a more rapid entropy decay rate with increasing grid size, from 100 m to 3000 m. This accelerated decrease in entropy for Guangzhou could be attributed to the city’s unique urban infrastructure variability, which becomes more pronounced beyond the subdivisional level (as highlighted in Table 2 and Figure 10). As grid sizes expand, larger cells encompass a broader variety of Weibo check-in points, leading to a sharper decrease in entropy values. This phenomenon suggests that Guangzhou’s urban layout and the distribution of social media check-in points within it are more sensitive to changes in spatial scale compared with the other cities.

These findings offer valuable insights for urban planners and researchers by shedding light on the randomness of city dwellers’ movements across both micro and macro scales. Researchers can gain a better understanding of the rate at which entropy values shift with spatial scale size, which allows them to select the most suitable scale for their studies. For example, if an individual visits various points nearby, applying a larger spatial scale could significantly alter the entropy values, as these points may aggregate and appear as one. However, for someone traveling greater distances, a larger spatial scale might still accurately capture their travel locations. Figure 11 shows an example of this. In Figure 11a, check-in points near each other (indicated by the left box) are displayed at a 100 m scale. However, they converge as the scale size increases. At a 300 m scale, all check-in points amalgamate into a singular point since all adjacent points converge at the center of a grid cell, as demonstrated in Figure 11d. Yet, the isolated check-in point in the right box remains unaffected by nearby points. Additionally, Figure 11 reveals that entropy values drop quickly as the scale size expands.

### 4.3. Limitations

In this study, we attempted to explore the effect of spatial scales on individual human movement patterns extracted from LBSM data. There are some limitations in this study. We used ROG and entropy in this study to identify the changes in the activity space of individuals for different scale sizes. We chose these two indicators because they perform well for sparse datasets like LBSM data [15], but it would be interesting to see how other activity space indicators (e.g., convex hull, alpha shape, standard deviation ellipse, etc.) are affected by different sizes of spatial scale. The dataset we utilized dates to 2015, which, because of time constraints and data unavailability, is not the most current dataset. However, it is sufficient to demonstrate the methodology and the impact of spatial scale on activity space modeling. Comparing the outcomes of this research with those derived from the most recent data can highlight how individual travel patterns have evolved over time. The temporal resolution of LBSM data can be concerning; however, one research goal of this paper is to explore how sparse datasets may be impacted by different analytical scales, so we believe the sparseness of the data does not undermine the validity of our findings. Instead, it provides a realistic challenge that urban planners and researchers must navigate when working with LBSM data, which is inherently intermittent and irregular in time.

Another limitation comes from how the dataset was processed. In this study, we opted to use raw pick-up numbers in each grid cell as it offers a straightforward, initial quantitative assessment of activities within specific areas. This method allows for the direct observation and comparison of pick-up frequencies across different urban zones without the complexity of additional processing. Future studies might explore the application of different density-based methods to adjust how the study area is divided, which could provide insights into the relative intensity of pick-ups when normalized by factors like area size or population density. This adjustment could highlight differences that raw counts alone might obscure, such as disproportionately high pick-ups in smaller or less populated areas, which offers a more refined understanding of spatial dynamics and needs [73].

In addition, Weibo data exhibit significant biases due to its user demographics, which are not representative of the general population. Younger individuals are overrepresented, skewing the age distribution and influencing the content and trends observed. Additionally, the geographic distribution of users is uneven, with urban areas, particularly major cities, being overrepresented compared with rural regions [74]. This creates a biased view of public opinion and behavior. Researchers can address these biases by implementing stratified sampling, weighting techniques, and cross-referencing with more representative data sources to ensure more accurate and generalizable findings [75]. In addition, there is a limitation in the geographical context. Our study is focused on only three cities in China, and the results may not be directly transferable to other contexts with different urban layouts, cultural habits, or technological penetration. Cross-cultural and cross-regional comparisons would be beneficial to understand the universality of the observed patterns. Other LBSM data sources (other than Weibo) can also be used to verify the findings of this study.

Lastly, the implications for urban infrastructure from the results are just hypotheses, which require further validation by planners and future studies. It can be helpful for city planners and policymakers to see how changing infrastructure, transportation modes, and policies may affect the individual travel patterns in different cities. This is beyond the scope of this research, as our goal is to provide quantitative evidence of how activity space calculations can be impacted by different grid sizes.

## 5. Conclusions

This research makes both methodological and empirical contributions. Empirically, we measured the effect of different spatial scales on individual activity spaces captured from LBSM data and examined the quantitative differences among the three cities. Methodologically, we applied different sizes of regular-shaped grids to analyze the following indicators of activity space: an external activity space indicator, the radius of gyration (ROG), and an internal activity space indicator, entropy. These indicators were chosen based on previous studies demonstrating their efficiency in analyzing sparse datasets like LBSM data. The methodology builds upon current analytical approaches by providing an analytical framework for exploring and comprehending the intricate aspects of activity spaces in urban settings.

To observe changes in these activity space indicators, we employed the following distinct ranges of scales: a fine scale, from 10 to 100 m, and a coarse scale, from 100 to 3000 m. We found that the ROG values increase but the entropy values decrease with the increase in the size of the spatial scale (grid size). This occurs because the locations of the original check-in points move far away from each other with the increase in grid sizes. As the grid size becomes larger, the middle points of the grid cells move far away from each other. On the other hand, as the grid size increases, it captures more check-in points within one grid cell, all the points captured by one grid cell are relocated to a single point (i.e., the middle of that cell), and thus, the probability of a user posting from a different location may decrease.

Moreover, the change in the ROG values with the increase in scale (grid) size for both fine and coarse scale ranges fit well into a linear regression model. On the other hand, the change in the entropy values with the increase in scale (grid) size for both fine and coarse scale ranges fit well into an exponential decay model. From the analysis of the parameters of both models, we find that different geographical features present in the study area may have a significant impact on the overall uniformity of urban areas and therefore impact how activity space indicators change with different spatial scales. For example, the Pearl River flows through various parts of Guangzhou city, disrupting the uniformity of urban structures in geographic space. Although the river passes through highly developed areas, there is minimal development directly over the water body. Therefore, very few people are present on the river, resulting in scarce social media check-ins from these locations. Thus, geographic features can lead to low population density in certain city parts, even if they are situated within high-density areas. The specific urban structure of a city will likely impact how activity spaces change with different grid sizes.

All in all, there is no perfect scale or grid size. The results of this study can be helpful for future researchers to determine how grid sizes correlate with activity space indicators, and therefore choose the appropriate spatial scale of their study depending on the research goal. A smaller spatial scale (grid size) will provide much more detailed information as the check-in points can be captured close to their original location, but it will be more computationally expensive and may cause inflated entropy values. As discussed in the Limitations Section, in the future, it would be interesting to investigate the effects of spatial scale on individual movements in urban areas by using other internal and external activity space indicators not covered in this study. In this study, we aggregated the LBSM dataset using regular-sized grid cells, a common method in urban-oriented research, as outlined in Section 2. Future studies might consider comparing different density-based methods and irregularly shaped sub-regions to further enhance our understanding. Moreover, the dataset used in this study is a few years old. It would be interesting to see the results of this study if performed again in the future using more recent datasets. A comparative study can be performed with a more recent dataset by using similar methods described in this research and comparing the results. Additionally, ROG and entropy can be calculated from different corners of the study area in future studies to determine how the results change. The models used in this study can be validated in future studies by using them to analyze different geospatial big datasets in different study areas. Finally, more accurate results for a specific study area can be obtained by combining the data with other geospatial big datasets (e.g., floating car data, mobile phone data, etc.) available in the same study area.

## Figures and Tables

**Figure 1 sensors-24-03796-f001:**
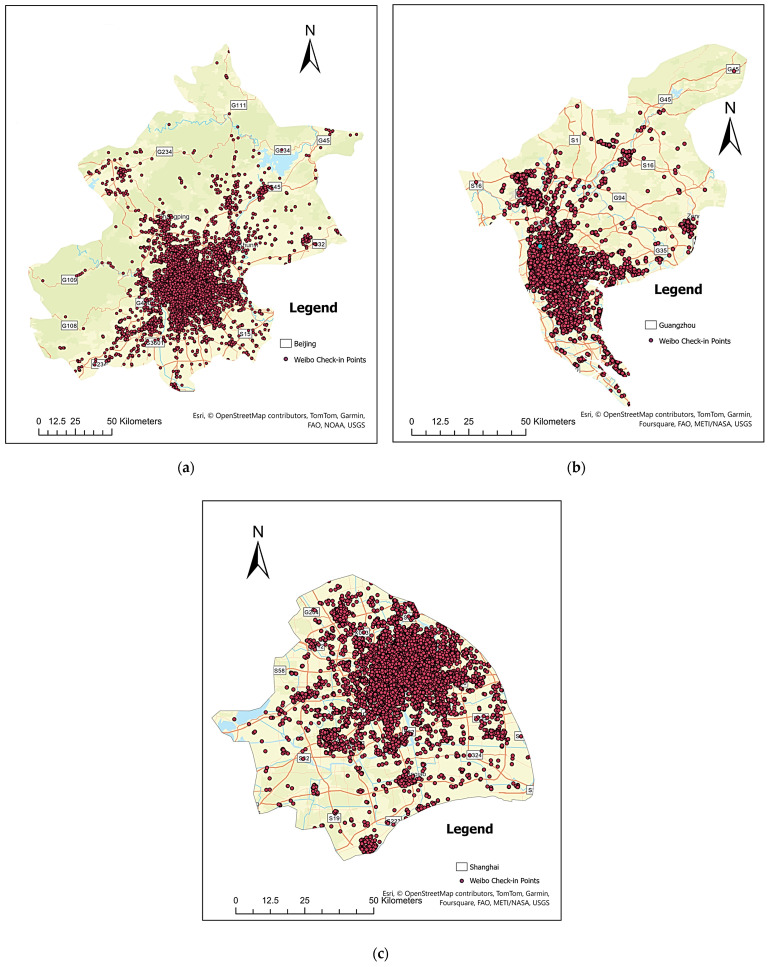
Distribution of Weibo check-in points in three different cities in China. (**a**) Beijing, (**b**) Guangzhou, and (**c**) Shanghai.

**Figure 2 sensors-24-03796-f002:**
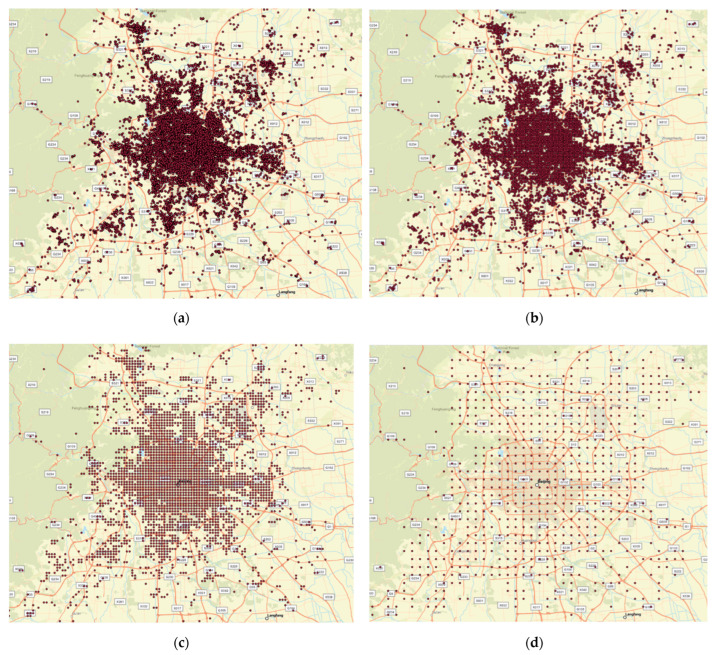
Weibo check-in points in Beijing, China, with different grid sizes. (**a**) Grid size of 100 m, (**b**) grid size of 500 m, (**c**) grid size of 1000 m, and **(d**) grid size of 3000 m.

**Figure 3 sensors-24-03796-f003:**
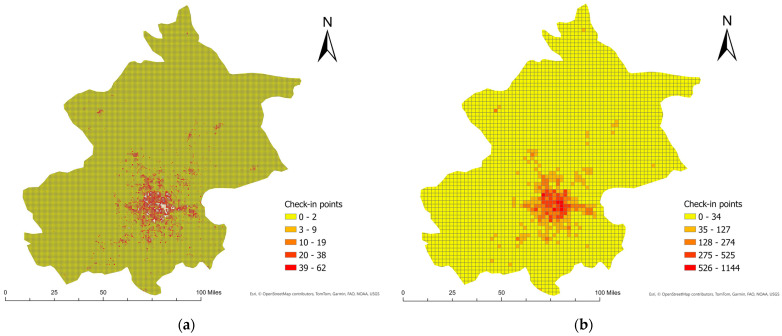
Number of check-in points aggregated within a grid cell in Beijing. (**a**) Grid size of 1000 m and (**b**) grid size of 3000 m.

**Figure 4 sensors-24-03796-f004:**
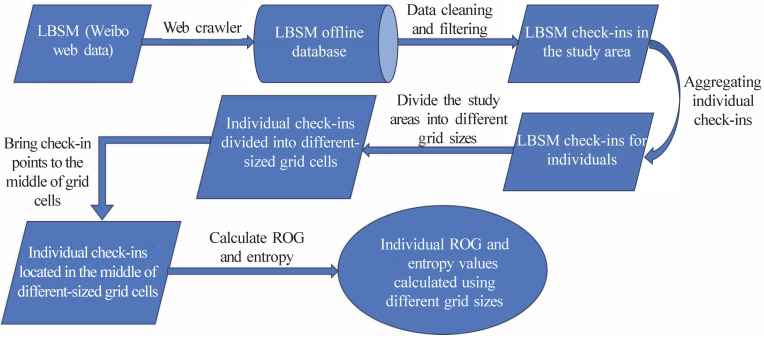
Flow chart of methods.

**Figure 5 sensors-24-03796-f005:**
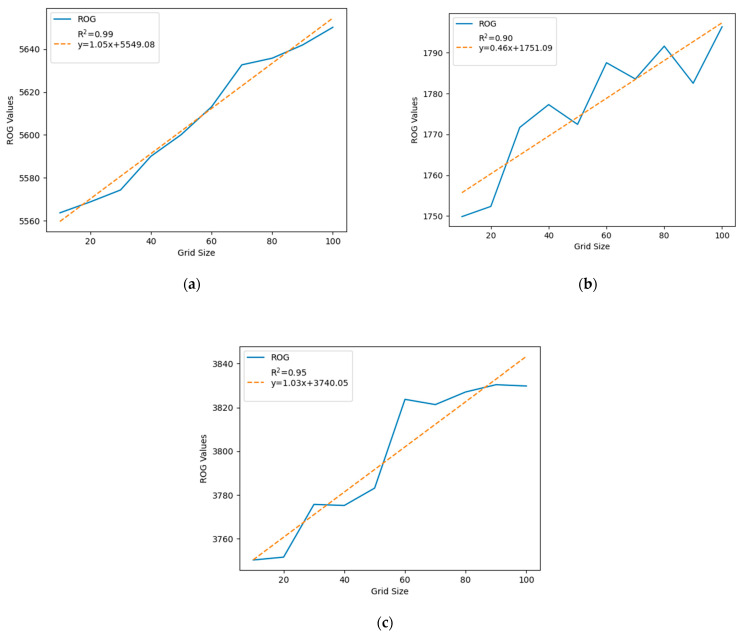
ROG values for grid sizes of 10 m to 100 m with an increment of 10 m for three different cities in China. (**a**) Beijing, (**b**) Guangzhou, and (**c**) Shanghai.

**Figure 6 sensors-24-03796-f006:**
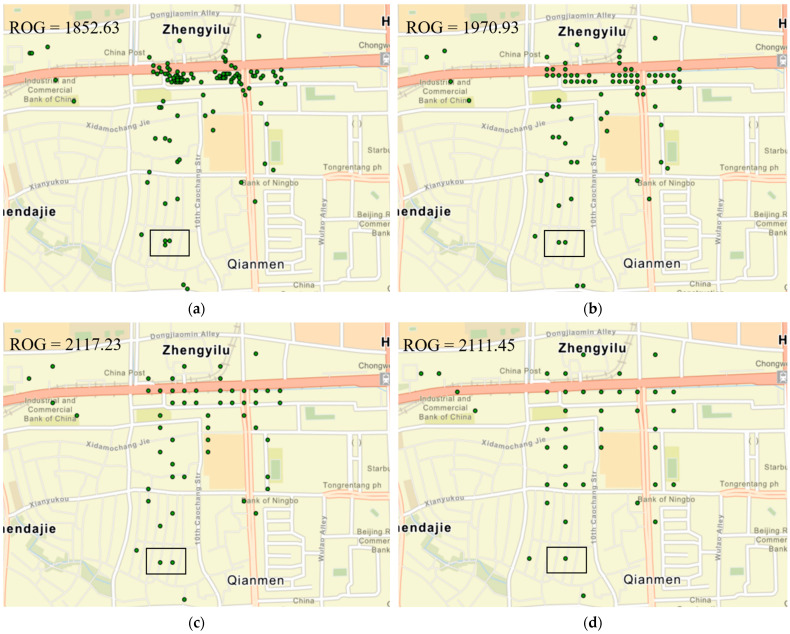
Change in Weibo check-in point distribution of an individual for different grid sizes in Beijing. (**a**) Grid size of 10 m, (**b**) grid size of 30 m, (**c**) grid size of 60 m, and (**d**) grid size of 90 m.

**Figure 7 sensors-24-03796-f007:**
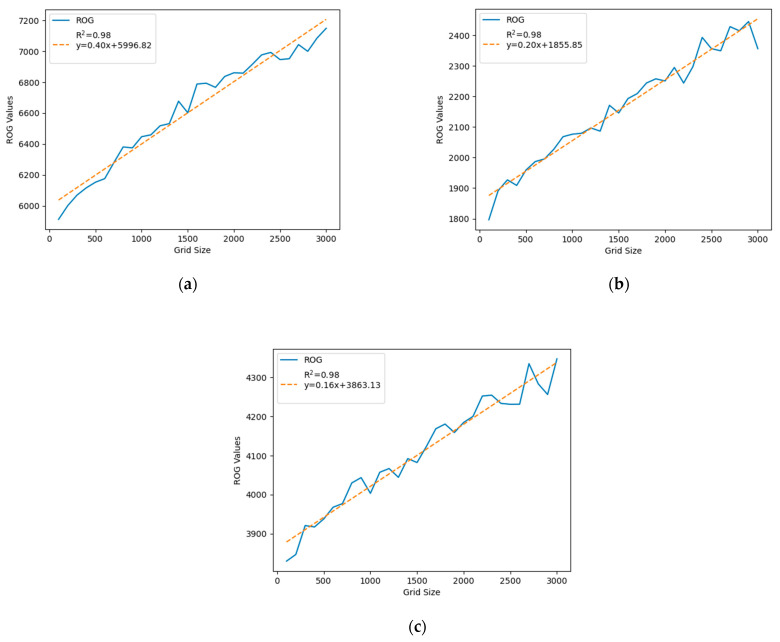
ROG values for grid sizes of 100 m to 3000 m with an increment of 100 m for three different cities in China. (**a**) Beijing, (**b**) Guangzhou, and (**c**) Shanghai.

**Figure 8 sensors-24-03796-f008:**
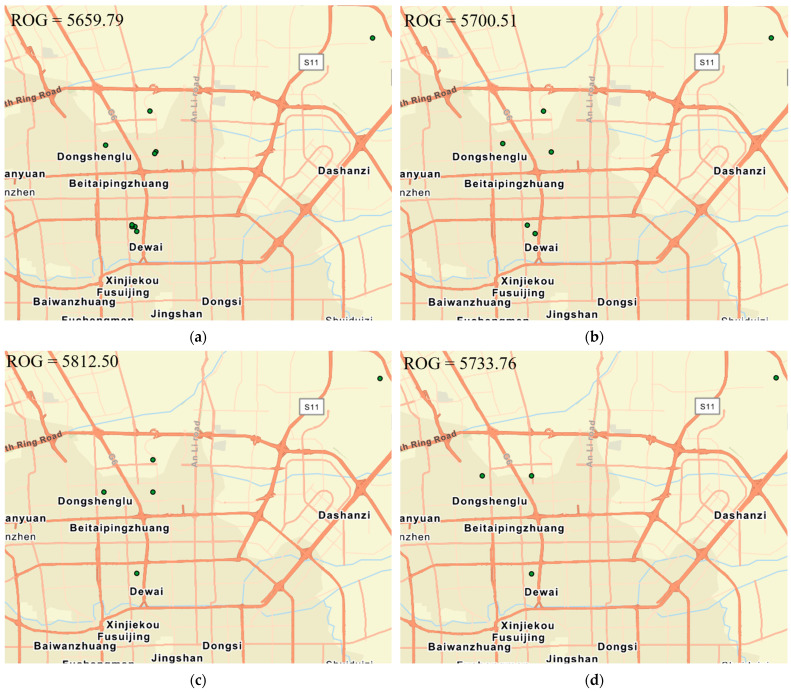
Change in Weibo check-in point distribution of an individual for different grid sizes in Beijing. (**a**) Grid size of 100 m, (**b**) grid size of 500 m, (**c**) grid size of 1000 m, and (**d**) grid size of 3000 m.

**Figure 9 sensors-24-03796-f009:**
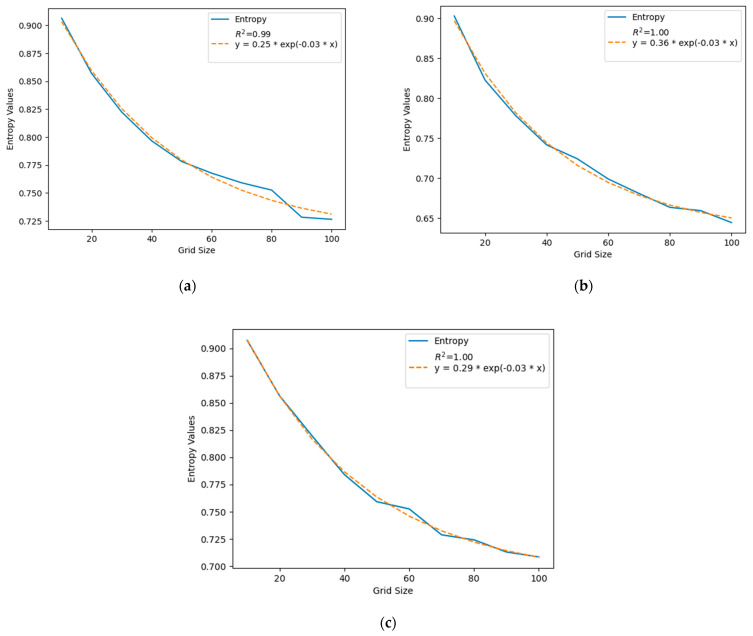
Entropy values for grid sizes of 10 m to 100 m with an increment of 10 m for three different cities in China. (**a**) Beijing, (**b**) Guangzhou, and (**c**) Shanghai.

**Figure 10 sensors-24-03796-f010:**
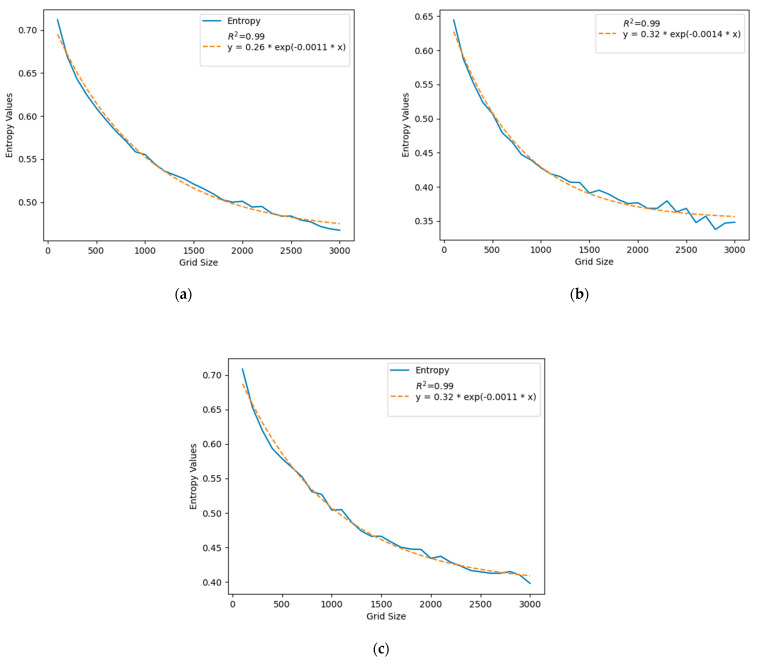
Entropy values for grid sizes of 100 m to 3000 m with an increment of 100 m for three different cities in China. (**a**) Beijing, (**b**) Guangzhou, and (**c**) Shanghai.

**Figure 11 sensors-24-03796-f011:**
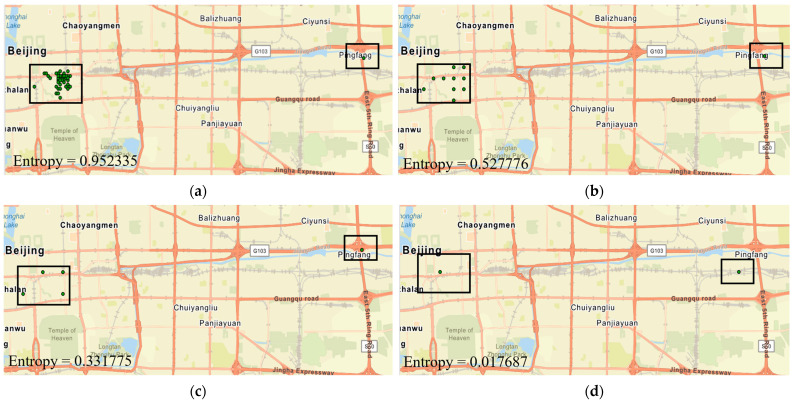
Weibo check-in points for a user at different spatial scales with entropy values: (**a**) 100 m, (**b**) 500 m, (**c**) 1000 m, and (**d**) 3000 m.

**Table 1 sensors-24-03796-t001:** Comparison of activity space indicators [16].

Activity Space Indicators	Pros	Cons
Minimum convex hull	Easy to calculate	Sensitive to outliers
Alpha shape	More accurate boundary than the convex hull and captures the detailed shape of an activity space	Sensitive to outliers
Standard deviation ellipse	Useful for characterizing the directional distributions of activity spaces (shape, orientation, etc.)	Does not capture the detailed shape of activity spaces
Radius of gyration	Less sensitive to outliers	Does not capture the detailed shape of activity spaces
Entropy	Less affected by the size and structure of a city	Cannot show the travel network of an individual’s visited locations
Minimum spanning tree	Capable of showing the complexity in the travel network formed by a user’s visited locations	Highly influenced by the scale and structure of a city

**Table 2 sensors-24-03796-t002:** Weibo metadata after data filtering and cleaning.

City	Number of Check-Ins (after Filtering)	Number of Users (with Check-Ins ≥ 10)
Beijing	50,355	2880
Guangzhou	33,122	1776
Shanghai	30,608	1639

**Table 3 sensors-24-03796-t003:** Explanation of the parameters in the exponential decay equation of the three studied cities for grid sizes of 10 m to 100 m with an increment of 10 m.

	Beijing	Guangzhou	Shanghai
*a*	0.25	0.36	0.29
*b*	0.03	0.03	0.03

**Table 4 sensors-24-03796-t004:** Explanation of the parameters in the exponential decay equation of the three studied cities for grid sizes of 100 m to 3000 m with an increment of 100 m.

	Beijing	Guangzhou	Shanghai
*a*	0.26	0.32	0.32
*b*	0.0011	0.0014	0.0011

## Data Availability

The datasets presented in this article are not readily available because of the data sharing policies of Weibo. Requests to access the datasets should be directed to http://www.weibo.com/.

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
