# Peer review of "The Impact of Scale on Extracting Individual Mobility Patterns from Location-Based Social Media"

_sensors, 2024, doi:10.3390/s24123796_

Round 1

Reviewer 1 Report

Comments and Suggestions for Authors

This paper explores how understanding human movement patterns in cities is essential for effective urban planning. In particular, these patterns were studied through surveys and the authors focused on the impact of spatial scale on analyzing big geospatial data from sources like mobile phones and social media. 

The study focuses on three Chinese cities (i.e., Beijing, Guangzhou, and Shanghai), using data gathered from Weibo to investigate how different spatial scales affect individual movement patterns. Two common indicators have been used, Radius of Gyration (ROG) and Entropy, to analyze activity spaces at fine (10-100 m) and coarse (100-3000 m) scales. 

The work is interesting, providing a methodological and empirical counterpoint. However, it is based on concepts widely used in the literature (ROG and entropy), so its novel aspects are quite limited. The application cases are limited to 3 cities in China, which present particular aspects of urban distribution and population density. It is not certain, therefore, that the same considerations apply to other cities in the world. Despite having an extensive number of bibliographical references, the introductory part of the work and the related works can be widely extended. In particular, it could be useful to provide the reader with useful references in the field of trajectory and sequential pattern mining from social media, focusing on the use of geolocalized data from social media (there is extensive literature on this).

In particular, I believe that the following aspects must be improved:

  • To understand users' mobility behaviors, we often try to reconstruct users' frequent trajectories as a sequence of longitude/latitude points, using sequential pattern or trajectory mining algorithms. Some related work about this topic are: 

    • [1] “Trajectory pattern mining. In Proceedings of the 13th ACM SIGKDD international conference on Knowledge discovery and data mining (pp. 330–339)”, 

    • [2] "Trajectory data mining: an overview. ACM Transactions on Intelligent Systems and Technology (TIST) 6.3 (2015): 1-41)”).

  • In addition, when evaluating user mobility behaviors, it is often difficult to make sense of such trajectories, as it is difficult to match a point of the trajectory with places of particular interest (e.g. monuments, squares, museums, etc.). For these reasons, these trajectories are often converted into sequences of regions of interest. About this aspect, you can refer to: 

    • [1] "G-RoI: Automatic Region-of-Interest detection driven by geotagged social media data". ACM Transactions on Knowledge Discovery from Data, vol. 12, n. 3, pp. 27:1-27:22, 2018.; 

    • [2] “Mining place-matching patterns from spatio-temporal trajectories using complex real-world places”. Expert Systems with Applications, 122 , 334-350.

  • Finally, using Big Data from social media makes the knowledge extraction process even more complex, since scalable, parallel, and distributed techniques are required for processing these large amounts of data. To this end you can refer to:

    • [1] “Big trajectory data mining: A survey of methods, applications, and services”. Sensors, 20(16), 4571

    • [2] "Knowledge discovery from large amounts of social media data." Applied Sciences 12.3 (2022): 1209.

    • [3] "Automatic detection of user trajectories from social media posts". Expert Systems with Applications, vol. 186, pp. 115733, 2021.

Other issues to address are:

  • In my opinion, displaying data on a map using a grid system with fixed size cells is not the best solution. Why not use density-based solutions (e.g. single-level or multilevel clustering)? The authors could argue about this for the sake of clarity. My feeling is that when the size of the grid becomes relatively large, the information reported loses its significance. In addition, information coming from clustering could be integrated with calculated metrics (ROG and entropy) to provide deeper and more useful insights?

  • Figure 3 should be more compact and each step of the flow chart should be discussed in the text.

  • In Section 3.1 the authors stated that 1.18 million georeferenced check-in points were collected, but in Table 1 about 100k check-ins are reported after cleaning. The cleaning process should be better described. In addition, what is the size of the final dataset? Can it be considered big data?

  • The data reported in Figures are not very explanatory, as when the scale changes, the grid does not show the number of checks that are aggregated into a single point. ROG and Entropy are calculated for each cell, but their average values are reported (e.g., in Figure 4). It could be useful to display this value for each cell? Or to use a color gradient visualization (such as heatmaps) to better understand the distribution of the values of these metrics across the different cells?

  • Fix typo at lines 407-408: “Where, y is the...” should stay on the same line.

Author Response

Please see the attached response file. Thank you.

Reviewer 2 Report

Comments and Suggestions for Authors

The paper deals with the impact of spatial scale on the analysis of human  movement patterns from  large geospatial data sources. In particular, authors study the effect of different spatial scales on individual human movement patterns. Two indicators are employed: the external activity space indicator "Radius of Gyration (ROG)"  and an internal activity space indicator "Entropy". The experimental evaluation is performed on several datasets.

The paper sounds interesting. However, there are several issues about its content and its structure that could be improved, as reported in the following:

-       Section II is very interesting and explainable about the previous approaches proposed in the literature. In my opinion, the section could be improved by adding a comparison table among the approaches proposed in literature, by identifying some features to make the comparison. For example, the rows can correspond to the approaches, while the columns can correspond to the features chosen for the comparison. This could make the comparison in a more complete treatment, by also describing how the different techniques achieve their specific goals. In addition, in my opinion there are several recent papers related to mobility-based applications that could be reported in Section II and cited in the paper:

- Epidemic forecasting based on mobility patterns: an approach and experimental evaluation on COVID-19 Data, 2022. Elsevier

- Big data analytics and smart cities: applications, challenges, and opportunities. 2023. Frontiers

- Big Data Analysis for Smart City Applications. Encyclopedia of Big Data Technologies. 2018. Springer

I think that these papers can add applicative examples to the paper treatment, and that can be added to manuscript bibliography.

-       The quality of figures should be improved. First, Figures 3 can be represented as an horizontal flow chart. The other figures (including 4, 5, 6, 10, etc.) could be  made by using more proper formats (smaller figures, on the same horizontal slot) and fonts. Please, use a similar template to have homogeneous formats for all figures in the. Also, I suggest to improve the readability and position (in the page) of Table 2. 

-       Some examples about the applicability of the analyzed techniques is necessary in the paper. For example, when the decay factor is used? How the adopted  measures are related to the decay factor?

-       On Figure 10 are represented squared areas related to mobility analysis. In my opinion, authors should add in the paper some treatment about how arbitrary shaped hotspots can be detected from mobility data. For example, there are several density-based clustering algorithms that can be used. Please, give a look at this paper: https://www.sciencedirect.com/science/article/abs/pii/S1574119222001018?dgcid=author, and get some ideas from it.

-       Overall, the paper has a very limited innovativeness. However, in my opinion, the suggestions could improve the quality of the paper and make it in a more complete treatment.

Comments on the Quality of English Language

it seems that quality of english is good, at least in average with other papers

Author Response

(The authors gave the same response as above.)

Reviewer 3 Report

Comments and Suggestions for Authors

This paper investigates the impacts of spatial scale on the analysis of human mobility patterns. The paper uses LBSM data collected in three cities in China and two indicators of the activity space, ROG, and Entropy, as an example to demonstrate the importance of considering spatial scales. Using 10-100m and 100-3000 m as two example ranges of distances, the comparison of results suggests that the ROG and Entropy values have contradicted trends with changes in spatial scales. Overall, the paper is well-organized and easy to follow.

Mobility data

-        Would each individual have the same number of days with valid records? Is the activity space of each individual derived for each individual person (for all time) OR each individual person-day? The reviewer asks because the activity space derived from multiple days tends to have more points and therefore different ROG and Entropy values even if the two individuals have similar mobility patterns. Intuitively, the impacts of spatial scales on these points would be more significant if we have more (or fewer?) days.

-        What is the original data precision level? Less than 10 meters?

-        If a person uses WeChat at the same location several times, will there be several points or just one point representing that activity? Some studies have used thresholds to reduce the “overestimates” of the place visit frequencies. Not sure if this would impact the analysis results in this paper.

Resampling

-        While increasing the grid size, there are many ways to create the “bigger” grid. For instance, if the original one has 10-meter interval and we have 0, 10, 20, 30, 40, 50, 60, …, for the x-axis. Then, when increasing it to 100, we could start with 0 and use 50 as the “center” of the grid OR we could start with -10 and use 40 as the “center” of the grid. This means 95 would belong to different grids in these two settings.

Another way to think about it is, whether the generatation of the larger grid is using the minX, minY (lower-left corner) of the spatial extent as the reference OR the maxX, maxY (upper-right) as the starting points. Since the spatial extent may not be evenly divided by 10, 100, …, the results would be different.

Presentation

-        The flow chart, figures, and maps can be further polished. The current style looks more like a semester paper rather than a journal publication. For instance, the flow chart should include some details regarding the processing settings (or demonstrations of processing results as graphs embedded); three subfigures in Figures 4, 6, and 8 be placed in one line, with larger fonts for axis labels.  

Comments on the Quality of English Language

Easy to follow. 

Author Response

Please see the attached response document. Thank you.

Reviewer 4 Report

Comments and Suggestions for Authors

This paper identifies the effect of different spatial scales on 17 individual human movement patterns as calculated from LBSM data. Generally speaking, this manuscript has a clear idea, reasonable structure and effective experiment, which is of certain reference significance for the study of individual behavior in cities.I think there are still a minor question that need to be added before publication.

There is a lot of bias in Weibo data. For example, the population structure is biased, the age of the population is biased, and the position of the sign-in crowd is biased. These questions are suggested to be quantified and partially explained in the discussion part of the article, which can make the conclusion more convincing.

Comments on the Quality of English Language

There is a problem with a small number of word orders.

Author Response

(The authors gave the same response as above.)

Round 2

Reviewer 1 Report

Comments and Suggestions for Authors

I am satisfied with the quality of the revised manuscript. The authors have addressed all the issues raised in the previous version of the paper, and now everything seems fine. For this reason, I recommend its acceptance.

Reviewer 2 Report

Comments and Suggestions for Authors

All issues have been addressed.